# PointDiT: Pixel-Space Diffusion for Monocular Geometry Estimation

**Haofei Xu** [1 2 3]  **Rundi Wu** [3]  **Philipp Henzler** [3]  **Nikolai Kalischek** [3]  **Michael Oechsle** [3]  **Fabian Manhardt** [3]
**Marc Pollefeys** [1 4]  **Andreas Geiger** [2 5]  **Federico Tombari** [3 6]  **Michael Niemeyer** [3]

## Abstract

State-of-the-art single-image 3D reconstruction methods often rely on complex hybrid architectures and loss functions (*e.g.*, MoGe), or necessitate compressing geometry into latent spaces (*e.g.*, GeometryCrafter) to leverage pre-trained latent diffusion models. In this work, we demonstrate that such architectural overhead and intricate loss formulations are unnecessary. We introduce a minimalist pixel-space Diffusion Transformer built on a plain ViT, which operates directly on raw 3D point map patches and is conditioned on image tokens from a pre-trained DINOv3. Unlike existing latent diffusion-based approaches, we train our diffusion backbone entirely from scratch, eliminating the need for point map tokenizers. We show that this streamlined approach yields results superior to complex latent-based diffusion models while remaining significantly simpler than hybrid alternatives. Notably, our model produces sharper geometric structures and achieves significantly better results on highly ambiguous regions, such as transparent objects.

## 1. Introduction

Monocular geometry estimation serves as a fundamental building block for 3D scene understanding, bridging the gap between 2D visual inputs and 3D spatial structures. In this work, we focus on predicting dense point maps from single RGB images (Wang et al., 2025b). Unlike depth maps, which capture only scalar distance and necessitate camera intrinsics to recover 3D structure, point maps represent the scene geometry directly in the camera coordinate system. This allows for immediate 3D reconstruction without knowledge of the camera's calibration parameters. However, mapping a single 2D image to a holistic 3D representation

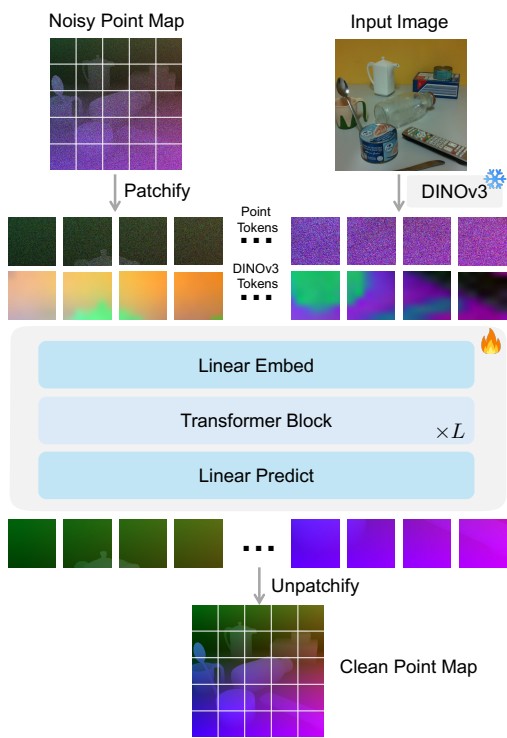

*Figure 1.* **Pixel-space diffusion on raw point map patches, conditioned on image tokens from a pre-trained DINOv3.** The 3D point map ($H \times W \times 3$) is visualized as an RGB image using color mapping to represent spatial $(X, Y, Z)$ coordinates.

remains an ill-posed problem due to the fundamental scale and depth ambiguities of perspective projection.

Existing approaches to this challenge can be broadly categorized into two distinct groups. The first category comprises deterministic feed-forward models ( (Yang et al., 2024; Bochkovskii et al., 2025; Piccinelli et al., 2025)). These methods often rely on complex hybrid architectures (Wang et al., 2025b;c;a), combining Vision Transformers (ViT) (Dosovitskiy, 2020) with convolutions (Ranftl et al., 2021), and require intricate loss functions (Wang et al., 2025b) to regularize training. Furthermore, due to the inherent ambiguity of the task, deterministic regressors tend to output the mean of the predictive distribution. This often results in over-smoothed geometry that lacks high frequency detail, particularly in complex scene regions (Figure 2).

---

[1]ETH Zurich [2]University of Tübingen, Tübingen AI Center [3]Google [4]Microsoft [5]KE:SAI [6]Technical University of Munich. Correspondence to: Haofei Xu <haofei.xu@inf.ethz.ch>.

*Proceedings of the 43rd International Conference on Machine Learning*, Seoul, South Korea. PMLR 306, 2026. Copyright 2026 by the author(s).

The second category attempts to resolve this ambiguity using Latent Diffusion Models (LDMs) (Rombach et al., 2022), such as GeometryCrafter (Xu et al., 2025b). While these methods leverage generative priors, they necessitate compressing point maps into a latent space via a Variational Autoencoder (VAE). Constructing a expressive latent space for geometric data is non-trivial and often requires sophisticated tokenizer designs. Consequently, these methods frequently suffer from information loss during the encoding-decoding process, struggling to reconstruct fine geometric structures accurately (Figure 2).

In this work, we demonstrate that such architectural overhead and intricate loss formulations are unnecessary. Inspired by the formulation of JiT (Li & He, 2026), we introduce a minimalist pixel-space diffusion framework that trains a diffusion model directly on the raw point map space. This allows us to leverage the probabilistic formulation of diffusion to model ambiguous regions without the signal degradation associated with VAEs. Our architecture is simple by design: a plain Vision Transformer (ViT) operating on point map patches. A critical component of our training strategy is the adoption of the $x$-prediction objective, i.e., predicting the clean point map directly, rather than the $v$-prediction target commonly used in flow matching (Salimans & Ho, 2022). Extending the findings of JiT (Li & He, 2026) beyond image generation, we demonstrate that this objective is highly effective for geometric data and yields significantly better results for point map estimation.

Our diffusion model is conditioned on the input RGB image to guide geometry prediction. While our model functions effectively with naive linear patchification (Dosovitskiy, 2020), we find that incorporating strong priors significantly enhances performance. Specifically, we leverage DINOv3 (Siméoni et al., 2025) as a robust general feature extractor. By conditioning our plain ViT backbone on DINOv3 tokens, we effectively bridge powerful priors from representation learning with diffusion training (Yu et al., 2025; Zheng et al., 2025).

We show that this streamlined approach yields results superior to complex latent-based diffusion models (Xu et al., 2025b) while remaining significantly simpler than hybrid deterministic alternatives (Wang et al., 2025b). In addition, our model excels in generating sharp geometric boundaries and resolving depth in highly ambiguous scenarios, such as transparent objects. Moreover, our PointDiT is able to achieve highly competitive results with just one-step diffusion, and the structural details can be further improved with additional diffusion sampling steps. Beyond this specific task, our work indicates that pixel-space diffusion can effectively generate non-pixel geometric data (*e.g.*, 3D point map in this paper), suggesting a simplified paradigm for future 3D and 4D generation tasks.

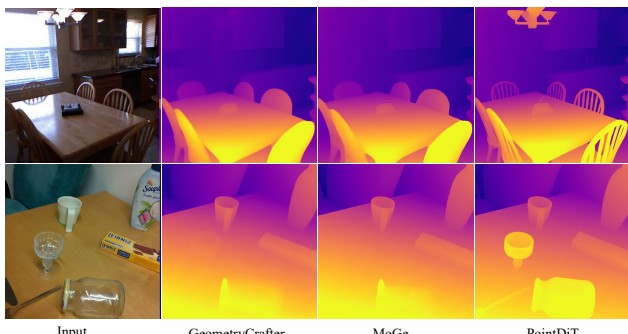

| Input | GeometryCrafter | MoGe | PointDiT |

*Figure 2.* **Depth comparisons.** We visualize the $z$-depth from the predicted 3D point maps. PointDiT is significantly better in terms of thin structures and transparent objects.

## 2. Related Work

**Latent Diffusion Models.** Latent Diffusion Models (LDMs) (Rombach et al., 2022) have become the dominant paradigm for high-resolution image synthesis, effectively decoupling the learning of semantic content from perceptual details by operating in a compressed latent space. Following this success, recent works have attempted to adapt LDMs for geometric tasks (Ke et al., 2024; He et al., 2024). For instance, methods like GeometryCrafter (Xu et al., 2025b) and various generative depth estimators (Ke et al., 2024) leverage pre-trained Variational Autoencoders (VAEs) to encode geometric maps into latent tokens. While effective for reducing computational costs, this compression strategy is fundamentally lossy. Constructing a tokenizer that preserves the high-frequency precision required for 3D geometry is non-trivial. Standard image VAEs often smooth out fine structural details or introduce artifacts during decoding (Xu et al., 2025a). In contrast, our approach bypasses the latent space entirely (Li & He, 2026). We demonstrate that avoiding this architectural overhead and the associated information loss enables the reconstruction of significantly sharper geometric boundaries.

**Pixel-Space Diffusion Models.** Prior to the dominance of latent methods, diffusion models operated directly in pixel space (Ho et al., 2020). Recently, the Diffusion Transformer (DiT) (Peebles & Xie, 2023) demonstrated that replacing the standard U-Net backbone with a Vision Transformer (ViT) yields state-of-the-art performance in class-conditional image generation. Building on this, recent flow-matching frameworks, such as JiT (Li & He, 2026), have shown that simplifying the training objective to direct data prediction ($x$-prediction) can significantly improve results. While these advancements have primarily focused on 2D image synthesis, we extend this minimalist philosophy to dense 3D geometry estimation. By treating 3D point maps as multi-channel images and training a plain ViT backbone from scratch conditioned on pre-trained DINOv3 feature embeddings, we

show that pixel-space diffusion is not only computationally feasible for 3D geometry but superior to complex latent alternatives in resolving ambiguities.

**Representation Learning and Generative Models.** There has been interesting study recently on connecting representation learning with generative models. REPA (Yu et al., 2025) and VA-VAE (Yao et al., 2025) observe that the performance of generative diffusion models can be improved dramatically by leveraging pre-trained vision encoders to regularize the latent space of diffusion models. RAE (Zheng et al., 2025) replaces the VAE in latent diffusion models with pretrained representation autoencoders (*e.g.*, DINOv2). From the high level, our PointDiT also shares similarity with RAE by drawing connection between DINOv3 and diffusion models. However, there are several key differences. First, RAE needs to be trained in two stages (reconstruction decoder and diffusion), while PointDiT is end to end. Second, RAE uses $v$-prediction, which requires scaling up the Transformer width, while PointDiT uses $x$-prediction and we successfully trained a smaller model variant PointDiT-B. Third, RAE requires 50 sampling steps for image synthesis, while PointDiT is able to perform one-step or few-step diffusion generation.

**Monocular Depth Estimation.** Estimating dense geometry from a single image is a longstanding problem in computer vision. Traditional discriminative approaches treated this as a regression task, utilizing Convolutional Neural Networks (CNNs) (Eigen et al., 2014) or, more recently, Transformers like DPT (Ranftl et al., 2021) and Depth Anything (Yang et al., 2024) to predict scalar depth maps. However, depth maps are 2.5D representations that require known camera intrinsics to be lifted into 3D space, which is often unavailable in unconstrained settings. Furthermore, recent generative approaches such as Marigold (Ke et al., 2024) repurpose pre-trained image diffusion models (e.g., Stable Diffusion) for depth estimation. While these methods leverage strong generative priors, they remain bound by the limitations of image VAEs. More recently, a pixel-space diffusion model PPD (Xu et al., 2025a) was proposed for monocular depth estimation. However, PPD still uses the $v$-prediction target, which performs significantly worse than $x$-prediction under our controlled comparisons (Table 4).

**Monocular Point Map Estimation.** To overcome the limitations of scalar depth, point map estimation predicts dense 3D coordinates $xyz$ directly in the camera coordinate system, enabling holistic 3D reconstruction without intrinsic calibration. The current state-of-the-art in this domain is represented by deterministic feed-forward models such as MoGe (Wang et al., 2025b). These methods typically employ complex hybrid architectures that fuse ViTs with convolutional layers and rely on intricate loss functions to enforce

geometric consistency. Despite their efficacy, deterministic regressors suffer from the inherent ambiguity of monocular projection, tending to output the mean of the predictive distribution. This often results in over-smoothed geometry, particularly in regions with high uncertainty or transparency. Our work addresses this by casting point map estimation as a probabilistic generation task, allowing our model to capture sharp, high-frequency details that deterministic baselines fail to resolve.

## 3. Approach

We address the problem of dense point map prediction from a single RGB image. Formally, given an input image $\mathbf{c} \in \mathbb{R}^{H \times W \times 3}$, our goal is to estimate a corresponding point map $\mathbf{x} \in \mathbb{R}^{H \times W \times 3}$, where each pixel encodes 3D spatial $xyz$ coordinates. To model the inherent ambiguities of 3D point map estimation from a single image, we propose a flow matching framework parameterized by a Vision Transformer (ViT) (Dosovitskiy, 2020; Peebles & Xie, 2023). Our method learns to transport a simple Gaussian noise distribution to the complex data distribution of point maps, conditioned on the input image.

### 3.1. Point Map Diffusion with Flow Matching

**Flow Matching.** We adopt the flow matching formulation to model the single image to point map diffusion process. Flow matching learns an Ordinary Differential Equation (ODE) that continuously transforms a prior noise distribution $p_0$ to the data distribution $p_1$.

Let $\mathbf{z}_t$ denote the state at time $t \in [0, 1]$, which is defined based on a linear interpolation path between a noise sample $\boldsymbol{\epsilon} \sim \mathcal{N}(\mathbf{0}, \mathbf{I})$ and a ground truth data sample $\mathbf{x} \sim p_{data}$:

$$\mathbf{z}_t = t \cdot \mathbf{x} + (1 - t) \cdot \boldsymbol{\epsilon}. \tag{1}$$

In this formulation, $t = 0$ corresponds to pure noise ($\mathbf{z}_0 = \boldsymbol{\epsilon}$) and $t = 1$ corresponds to the clean data ($\mathbf{z}_1 = \mathbf{x}$). The vector field $\mathbf{v}_t(\mathbf{z}_t)$ generating this probability path is defined as the time derivative of the state:

$$\mathbf{v}_t = \frac{d\mathbf{z}_t}{dt} = \mathbf{x} - \boldsymbol{\epsilon}. \tag{2}$$

This linear path induces a constant velocity vector for each sample pair $(\mathbf{x}, \boldsymbol{\epsilon})$, ensuring a direct and straight transport between the noise and data distributions.

**Image Conditioned Flow Matching.** In this paper, we extend this framework to model the conditional distribution $p(\mathbf{x}|\mathbf{c})$, where $\mathbf{c}$ is the input RGB image and $\mathbf{x}$ is the target dense point map. Consequently, we aim to learn a conditional vector field $\mathbf{v}_\theta(\mathbf{z}_t, t|\mathbf{c})$ that predicts the target velocity derived in Equation (2). By conditioning on $\mathbf{c}$, the model exploits the image's spatial context to disambiguate

the noise, ensuring the vector field accurately directs the flow toward the target point map geometry.

**Point Map Normalization.** Unlike standard RGB images bounded within $[0, 1]$, dense point maps exhibit varying coordinate ranges depending on the scene domain (e.g., indoor environments vs. outdoor landscapes). In our flow matching formulation, the training target relies on the interpolation $\mathbf{z}_t = t\mathbf{x} + (1 - t)\epsilon$, where the noise $\epsilon$ follows a fixed standard normal distribution $\mathcal{N}(\mathbf{0}, \mathbf{I})$. If the scale of the point data $\mathbf{x}$ significantly exceeds that of the noise, the data signal dominates the interpolation path even at near-zero time steps. This prevents the noise from effectively destroying the data structure, which destabilizes diffusion training. To mitigate this, we standardize the point maps prior to training. For each point map, we compute the centroid $\boldsymbol{\mu}$ and a scalar scale factor $s$, defined as the mean Euclidean distance of the points from the centroid. The normalized data $\tilde{\mathbf{x}}$ is given by:

$$\tilde{\mathbf{x}} = \frac{\mathbf{x} - \boldsymbol{\mu}}{s}. \tag{3}$$

This normalization ensures the data distribution has a scale comparable to the noise prior, facilitating stable flow matching. Our model is trained in this normalized space, and thus our point map prediction is affine-invariant (with unknown scale and shift parameters).

**Sky Processing.** To accommodate the effectively infinite depth of the sky in outdoor scenes, we project sky points onto a virtual, far-plane sphere with a fixed radius of 3 (corresponding to $3\sigma$ of the standard normal noise distribution). During training, we assign a lower loss weight (0.01) to these sky points compared to non-sky regions (1.0). This reduced weighting penalizes errors in the sky region less harshly, effectively embedding the geometric prior that the sky is a distant background element without allowing its arbitrary pseudo-depth values to dominate the optimization. Consequently, this enables stable joint training across heterogeneous indoor and outdoor datasets.

### 3.2. Architecture

We parameterize the conditional flow matching using a Vision Transformer (ViT) $F_\theta$. The network accepts the noisy point map $\mathbf{z}_t$, the current time step $t$, and the conditioning image $\mathbf{c}$ as inputs, and uses the clean point map as the prediction target. This is a crucial difference with previous flow matching models which usually predict velocity. This is inspired by the JiT (Li & He, 2026) model for image generation, and we demonstrate that the clean data prediction target is also crucial for the point map data. Figure 1 shows an overview of our architecture.

**Point Map Patchification.** The noisy point map input is a tensor $\mathbf{z}_t \in \mathbb{R}^{H \times W \times 3}$. To prepare this dense map for the ViT, we first partition it into a regular grid of non-overlapping patches. Using a patch size of $p \times p$, this operation yields $N = (H/p) \times (W/p)$ distinct patches. Each patch is flattened into a vector of size $3p^2$. These flattened patches are then mapped to the embedding dimension $D$ via a learnable linear projection layer $\phi$:

$$\mathbf{T}_z = \phi(\text{Reshape}(\mathbf{z}_t)) \in \mathbb{R}^{N \times D}. \tag{4}$$

Here, the Reshape operation converts the input tensor from $\mathbb{R}^{H \times W \times 3}$ to a sequence of flattened patches in $\mathbb{R}^{N \times 3p^2}$, and $\phi$ applies a linear transformation $\mathbb{R}^{3p^2 \to D}$.

**Image Conditioning.** The conditioning image $\mathbf{c}$ provides the static structural guidance for the generation process. While this input could be processed via standard learnable patch embeddings, we instead exploit the clean nature of $\mathbf{c}$ to leverage powerful pre-trained representations. Specifically, we employ a frozen DINOv3 encoder (Siméoni et al., 2025) to extract patch tokens. Unlike RAE (Zheng et al., 2025) that only uses the last layer, we found it is beneficial to combine DINOv3 features at different layers. In particular, we use four uniformly spaced intermediate layers, which follows the popular layer selection of the DPT (Ranftl et al., 2021) head. Unlike DPT that relies on sophisticated convolutions to fuse these features, we simply concatenate these tokens along the channel dimension to capture a rich hierarchy of features, ranging from low-level details to high-level abstractions. This operation yields a composite image representation $\mathbf{T}_c \in \mathbb{R}^{N \times 4D}$, where $D$ corresponds to the feature dimension. To ensure structural alignment, we employ an identical patch size $p = 16$ and embedding dimension $D$ for both the point map and DINOv3 branches.

**Image and Point Map Fusion.** Given the spatial alignment between $\mathbf{T}_c$ and $\mathbf{T}_z$, we fuse the modalities via channel-wise concatenation. The input to the Transformer backbone is:

$$\mathbf{T}_{in} = \text{Concat}(\mathbf{T}_c, \mathbf{T}_z) \in \mathbb{R}^{N \times 5D}. \tag{5}$$

A linear embedding layer projects this concatenated representation to the model dimension $D$. The sequence is then processed by a stack of standard Transformer blocks consisting of Multi-Head Self-Attention and MLP layers.

**Clean Point Map Prediction.** The Transformer backbone outputs a sequence of refined tokens $\mathbf{T}_{out} \in \mathbb{R}^{N \times D}$. To recover the dense point map, we first apply a linear prediction head that projects each token from dimension $D$ back to the flattened patch dimension $3p^2$. This yields a sequence of patch vectors in $\mathbb{R}^{N \times 3p^2}$. We then apply an *unpatchification* operation, which rearranges these vectors back into the

original spatial grid structure $(H/p) \times (W/p) \times p \times p \times 3$ and permutes dimensions to reconstruct the full resolution tensor $\hat{\mathbf{x}} \in \mathbb{R}^{H \times W \times 3}$. This output $\hat{\mathbf{x}}$ represents the model's estimate of the clean point map at the current step.

### 3.3. Training

**Noise Schedule.** To sample the time step $t \in [0, 1]$ during training, we employ a logit-normal (Esser et al., 2024) distribution following JiT (Li & He, 2026). Formally, we sample a variable $z \sim \mathcal{N}(\mu, \sigma^2)$ and map it to the time domain via the sigmoid function, $t = \sigma(z)$.

In our point map diffusion task, we observe that the asymptotic boundaries of the sigmoid schedule preclude the model from training on exact pure noise states ($t = 0$). This creates a train-test discrepancy, as inference is strictly initialized at $t = 0$. Consequently, the model may struggle to initiate the flow trajectory correctly from the prior (Lin et al., 2024). To resolve this, we introduce a rectified sampling strategy: with a probability of $p_{\text{zero}} = 0.1$, we explicitly set the sampled time step to $t = 0$, overriding the logit-normal sample. This ensures the model is well-calibrated to the pure noise distribution encountered at the start of inference.

**Training Objective.** While our network $F_\theta$ is parameterized to predict the clean point map $\hat{\mathbf{x}}$, we optimize the model in the velocity space following JiT (Li & He, 2026).

**Velocity Loss.** During training, we sample a time step $t \sim \mathcal{U}(0, 1)$ and noise $\boldsymbol{\epsilon} \sim \mathcal{N}(\mathbf{0}, \mathbf{I})$ to construct the latent input $\mathbf{z}_t$. We first obtain the clean data prediction $\hat{\mathbf{x}} = F_\theta(\mathbf{z}_t, t, \mathbf{c})$ and convert it into an estimated velocity $\hat{\mathbf{v}}_t$ using the algebraic relationship derived from the interpolation path (Equation (1)):

$$\hat{\mathbf{v}}_t(\mathbf{z}_t, t) = \frac{\hat{\mathbf{x}} - \mathbf{z}_t}{1 - t}. \qquad (6)$$

To ensure numerical stability as $t \to 1$, we clip the denominator $(1 - t)$ to a minimum threshold $\delta = 0.05$ following JiT (Li & He, 2026).

We minimize the Mean Squared Error (MSE) between this estimated velocity $\hat{\mathbf{v}}_t$ and the constant ground truth velocity target $\mathbf{u}_t = \mathbf{x} - \boldsymbol{\epsilon}$:

$$\mathcal{L}_{\text{denoise}} = \mathbb{E}_{\mathbf{x}, t, \epsilon} \left[ \| \hat{\mathbf{v}}_t - (\mathbf{x} - \boldsymbol{\epsilon}) \|_2^2 \right]. \qquad (7)$$

**Relative Point Loss.** Point maps often exhibit a high dynamic range, where distant points posses large coordinate norms that dominate standard error metrics. Training the model directly in pixel space without autoencoders allows us to easily add additional regularization on points. To ensure the model accurately reconstructs local details in near regions, we introduce a relative loss term on the predicted

clean data $\hat{\mathbf{x}}$:

$$\mathcal{L}_{\text{rel}} = \mathbb{E}_{\mathbf{x}} \left[ \frac{1}{M} \sum_{i=1}^{M} \frac{\| \hat{\mathbf{x}}_i - \mathbf{x}_i \|_1}{\| \mathbf{x}_i \|_2 + \xi} \right], \qquad (8)$$

where $i$ indexes the pixels, $M$ is the total pixel count, and $\xi$ is a small stability constant.

**Total Loss.** The final optimization objective is the weighted sum:

$$\mathcal{L} = \mathcal{L}_{\text{denoise}} + \lambda \mathcal{L}_{\text{rel}}, \qquad (9)$$

where $\lambda = 0.1$ is the loss weight. We train our full model end-to-end.

### 3.4. Inference

During inference, we recover $\mathbf{x}$ from pure noise $\mathbf{z}_0 \sim \mathcal{N}(\mathbf{0}, \mathbf{I})$ conditioned on an image $\mathbf{c}$ by solving the ODE $d\mathbf{z}_t = \mathbf{v}_\theta(\mathbf{z}_t, t)dt$ from $t = 0$ to $t = 1$. We employ a standard Euler solver with step size $\Delta t$. At each step $t$, we predict the clean data $\hat{\mathbf{x}}$, derive the velocity $\hat{\mathbf{v}}_t$, and update the state:

$$\mathbf{z}_{t+\Delta t} \leftarrow \mathbf{z}_t + \Delta t \cdot \hat{\mathbf{v}}_t. \qquad (10)$$

This iterative process transports the sample along the learned linear trajectory to reconstruct the final point map.

Surprisingly, we observe that our model is able to do single-step feed-forward inference and achieve competitive results, and we can use more steps to further improve the results with the same model. Thus it's efficient during inference time. Interestingly, this behavior has also been observed by (Garcia et al., 2025) in diffusion-based depth estimation models, probably because the model was trained to be fairly robust to different noise inputs or even constant zeros.

## 4. Experiments

### 4.1. Datasets

We employ a two-stage training strategy to facilitate efficient training process. Our model is first pre-trained on images at $256 \times 256$ resolution and subsequently fine-tuned at $512 \times 512$ resolution. Our model is trained exclusively on synthetic datasets. For the initial pre-training stage, we use the synthetic SceneNet-RGBD (McCormac et al., 2017) dataset, which provides approximately 5.36 million photo-realistic RGB-D samples. To align the data with our 3D representation, we convert the raw depth maps into point maps using the provided camera intrinsics. The fine-tuning stage leverages a high-fidelity mixture of 11 synthetic datasets: Hypersim (Roberts et al., 2021), VKITTI2 (Cabon et al., 2020), UrbanSyn (Gómez et al., 2025), Synscapes (Wrenninge & Unger, 2018), TartanAir (Wang et al., 2020),

OmniWorldGame (Zhou et al., 2025), EDEN (Lê et al., 2021), IRS (Wang et al., 2019), Dynamic Replica (Karaev et al., 2023), MVSSynth (Huang et al., 2018), and TartanAirV2 (Wang et al., 2025d). We exclusively use synthetic data for training for two primary reasons: 1) Geometric Precision: Synthetic environments provide "pixel-perfect" ground-truth point maps, which are essential for our model's objective of learning dense 3D distributions. 2) Domain Agnosticism: Since our architecture aims to model the underlying geometry (point maps) rather than low-level image textures, the appearance gap is of secondary importance. To further bridge the domain gap between synthetic and real-world distributions, we incorporate frozen features from a pre-trained DINOv3 backbone. These self-supervised representations provide robust, domain-invariant visual cues that allow our model to focus on geometric reconstruction while maintaining generalizability to natural images.

### 4.2. Implementation Details

**Model Configurations.** We implement three scaling variants of our architecture: PointDiT-B (Base), PointDiT-L (Large), and PointDiT-H (Huge), following the configuration standards established by JiT (Li & He, 2026). For the visual backbone, we utilize frozen DINOv3 features to extract patch-level embeddings. The capacity of the DINOv3 backbone is scaled in correspondence with each variant (e.g., ViT-L features for PointDiT-L) to maintain representative alignment. Apart from the frozen DINOv3 encoder, all transformer layers and prediction heads are trained from scratch. We use the same patch size 16 for all the model variants.

**Training Schedule.** Our training curriculum consists of a large-scale pre-training phase followed by high-resolution fine-tuning. We adopt the AdamW optimizer with a learning rate schedule and hyperparameters consistent with the JiT framework.

During the $256 \times 256$ pre-training phase, all variants (PointDiT-B, L, and H) are trained for 30 epochs (including a 5-epoch warmup) on $16 \times$ H100 GPUs, requiring between 16 and 22 hours depending on model capacity. The $512 \times 512$ fine-tuning phase uses more GPUs to manage the increased resolution: PointDiT-B and PointDiT-L utilize $64 \times$ H100 GPUs for 8 and 5 epochs, respectively. PointDiT-H is fine-tuned for 3 epochs on $128 \times$ H100 GPUs. Interestingly, we observe that the larger models converges faster and they don't need to train for many epochs.

### 4.3. Evaluation Setup and Metrics

To assess the zero-shot generalization capabilities of our model in diverse practical scenarios, we evaluate on seven prominent real-world datasets: DIODE (Vasiljevic et al., 2019), KITTI (Geiger et al., 2012), NYUv2 (Silberman et al., 2012), ETH3D (Schöps et al., 2017), HAMMER (Jung et al., 2022), iBims-1 (Koch et al., 2018), and Booster (Zama Ramirez et al., 2022). These datasets represent a wide array of environments, ranging from indoor rooms to complex outdoor driving scenes. Consistent with our training curriculum, we evaluate performance at both $256 \times 256$ and $512 \times 512$ resolutions. Given the heterogeneous aspect ratios and resolutions of the original test sets, we adopt a standardized preprocessing pipeline. More specifically, the input images are first rescaled such that the shorter side (height) matches the target resolution (256 or 512 pixels), and then a center crop is subsequently applied to produce the final square input required by the model architecture. To ensure a rigorous and fair comparison, we benchmark our approach against several state-of-the-art baselines. We evaluate the publicly available pre-trained weights of these methods using the exact same preprocessing and cropping protocol described above.

Our model predicts affine-invariant point maps, from which affine-invariant depth maps can be derived by extracting the $z$-component of each point. For evaluation, we follow the alignment procedure in MoGe (Wang et al., 2025b), and determine the optimal scale and shift by solving a least-squares problem that minimizes the difference between the prediction and the GT. We evaluate the quality of our predictions across both point map and depth map domains using standard depth estimation metrics (Wang et al., 2025b):

- Accuracy ($\delta_1$): The percentage of pixels where the ratio between the prediction and GT is less than 1.25.

- Relative Absolute Error (Rel): Defined as $\frac{1}{N} \sum \frac{|y - \hat{y}|}{y}$, measuring the global scale-normalized error.

- Geometric Edge Fidelity (BF1): To specifically assess the model's ability to recover fine structural details and sharp depth discontinuities, we evaluate local boundary metrics following Depth Pro (Bochkovskii et al., 2025).

We evaluate both point cloud and depth estimation quality, and use Rel$^p$ and $\delta_1^p$ to denote point cloud metrics, and Rel$^d$ and $\delta_1^d$ to denote depth metrics.

### 4.4. Evaluation Results

We present comparison results in Table 1 and Figure 3.

**Single-Step Feed-Forward Inference.** While our model is trained with diffusion, we find that our architecture is able to perform single-step feed-forward inference, as also observed in (Garcia et al., 2025) on diffusion-based monocular depth estimation. In Table 2, we investigate the sensitivity of our model to the noise sampling process during inference in a single step. our model demonstrates significant robustness across various stochastic initializations. When com-

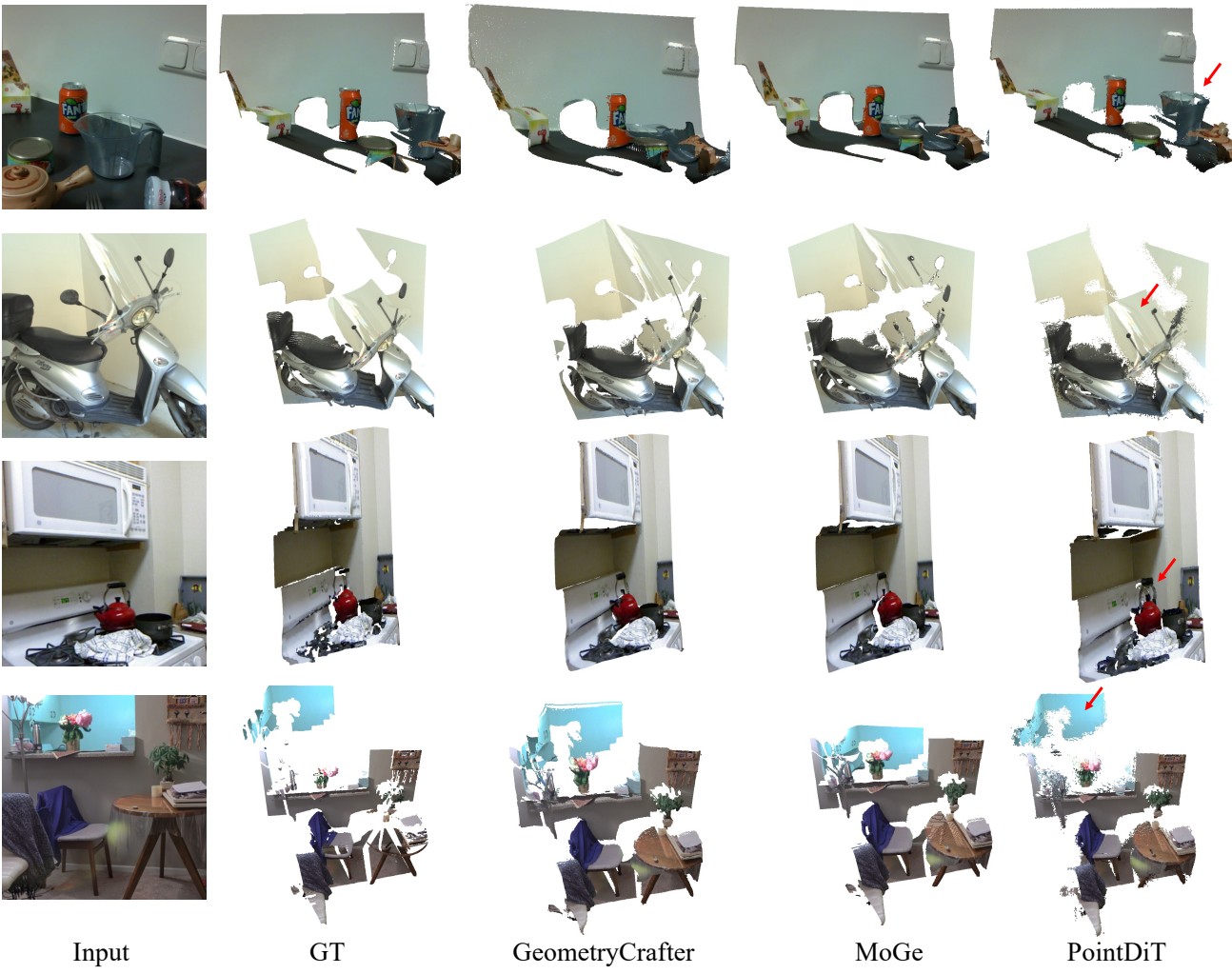

| Input | GT | GeometryCrafter | MoGe | PointDiT |
|---|---|---|---|---|

*Figure 3.* **Point map comparisons.** Our PointDiT is significantly better in terms of reconstructing transparent objects (1st and 2nd rows), thin structures (3rd row), and maintaining a more accurate relative scale across the global scene (4th row).

paring different random noise seeds, the performance fluctuations are negligible, with $\text{Rel}^p$ and $\delta_1^p$ remaining nearly constant. More notably, the model maintains high-fidelity predictions even under deterministic sampling, where the input noise is set entirely to zeros. This "all zeros" configuration not only matches but occasionally exceeds the performance of stochastic sampling. These results suggest that our model has successfully learned a robust mapping from the DINOv3-encoded image patch tokens to the geometric point map. Moreover, the single-step feed-forward inference significantly reduces computational overhead at test time while maintaining accuracy. In Table 1, we demonstrate that our single-step feed-forward variant PointDiT-H already outperforms previous methods, despite that our network architecture is just a plain ViT.

**Multi-Step to Improve Local Structures.** Thanks to our diffusion-based formulation, we are also able to benefit

from additional multi-step inference with the same weight-sharing architecture. In Table 1, we show that with more inference steps, the boundary metric BF1 gets gradually improved and significantly outperforms previous methods. The average metrics Rel and $\delta_1$ remain stable since our model already produces high-quality results with a single-step feed-forward inference. The visual results are shown in Figure 4. The ability of being able to perform different numbers of inference steps with the same network architecture demonstrates the flexibility of our model.

**Significantly Sharper Local Structures.** In Figure 3, we show the visual comparisons with previous methods. Our model significantly improves the sharpness of the local structures while maintaining the high quality of the overall geometry. In contrast to the latent-diffusion based method GeometryCrafter, our method does not suffer from the inherent lossy compression introduced by the VAE, as can be seen

*Table 1.* **Comparisons**. Average results on 7 real-world evaluation datasets with 3444 samples. The image resolution is $512 \times 512$. $Rel^p$ and $\delta_1^p$ are point map metrics, while $Rel^d$ and $\delta_1^d$ are depth map metrics. BF1 measures boundary sharpness.

| Method | $Rel^p \downarrow$ | $\delta_1^p \uparrow$ | $Rel^d \downarrow$ | $\delta_1^d \uparrow$ | BF1 $\uparrow$ | Param (M) | Time (ms) |
|---|---|---|---|---|---|---|---|
| GeometryCrafter | 5.45 | 96.75 | 3.52 | 97.84 | 4.64 | 1937 | 1178 |
| Depth Pro | 5.71 | 96.71 | 3.84 | 97.63 | 9.41 | 952 | 68 |
| UniDepthV2 | 4.45 | 97.35 | 2.86 | 98.52 | 6.94 | 354 | 26 |
| MoGe | **4.21** | 97.45 | 3.10 | 98.01 | 5.61 | 314 | 34 |
| MoGe-2 | 4.53 | 97.46 | 2.90 | 98.45 | 7.40 | 326 | 24 |
| PointDiT-B (1 step) | 5.84 | 96.71 | 3.70 | 97.84 | 8.18 | 223 | 31 |
| PointDiT-B (2 steps) | 5.81 | 96.77 | 3.64 | 97.86 | 8.88 | 223 | 47 |
| PointDiT-B (3 steps) | 5.83 | 96.79 | 3.64 | 97.86 | 9.09 | 223 | 63 |
| PointDiT-B (4 steps) | 5.85 | 96.80 | 3.64 | 97.86 | 9.16 | 223 | 79 |
| PointDiT-L (1 step) | 4.90 | 97.42 | 3.15 | 98.22 | 9.56 | 771 | 65 |
| PointDiT-L (2 steps) | 4.84 | 97.52 | 3.09 | 98.24 | 10.11 | 771 | 87 |
| PointDiT-L (3 steps) | 4.85 | 97.54 | 3.09 | 98.25 | 10.36 | 771 | 109 |
| PointDiT-L (4 steps) | 4.85 | 97.55 | 3.09 | 98.25 | **10.49** | 771 | 131 |
| PointDiT-H (1 step) | 4.45 | 97.93 | 2.81 | 98.51 | 9.79 | 1807 | 72 |
| PointDiT-H (2 steps) | 4.38 | 97.99 | **2.75** | **98.54** | 10.31 | 1807 | 116 |
| PointDiT-H (3 steps) | 4.39 | 98.01 | **2.75** | **98.54** | 10.44 | 1807 | 160 |
| PointDiT-H (4 steps) | 4.40 | **98.02** | **2.75** | **98.54** | 10.49 | 1807 | 204 |

*Table 2.* **Single-step feed-forward inference**. We start from random noises or deterministic all zeros.

| Method | $Rel^p \downarrow$ | $\delta_1^p \uparrow$ | $Rel^d \downarrow$ | $\delta_1^d \uparrow$ | BF1 $\uparrow$ |
|---|---|---|---|---|---|
| rand noise (seed 1) | 4.454 | 97.928 | 2.815 | 98.505 | 9.772 |
| rand noise (seed 2) | 4.452 | 97.938 | 2.811 | 98.513 | 9.778 |
| rand noise (seed 3) | 4.454 | 97.921 | 2.812 | 98.513 | 9.772 |
| all zeros (no rand) | 4.446 | 97.934 | 2.806 | 98.508 | 9.792 |

*Table 3.* **Diffusion vs. non-diffusion**. Non-diffusion: zero noise, zero timestep.

| Method | $Rel^p \downarrow$ | $\delta_1^p \uparrow$ | $Rel^d \downarrow$ | $\delta_1^d \uparrow$ | BF1 $\uparrow$ |
|---|---|---|---|---|---|
| Non-diffusion | 10.10 | 89.58 | 6.36 | 93.47 | 10.90 |
| Diffusion | **9.10** | **91.48** | **5.53** | **94.88** | **13.92** |

*Table 4.* **Prediction target**.

| Target | $Rel^p \downarrow$ | $\delta_1^p \uparrow$ | $Rel^d \downarrow$ | $\delta_1^d \uparrow$ | BF1 |
|---|---|---|---|---|---|
| $v$-pred | 35.44 | 30.03 | 24.07 | 58.21 | 0.46 |
| $x$-pred | **9.29** | **91.18** | **5.5** | **95.08** | **13.47** |

*Table 5.* **Image patch embedding**.

| Embedding | $Rel^p \downarrow$ | $\delta_1^p \uparrow$ | $Rel^d \downarrow$ | $\delta_1^d \uparrow$ | BF1 $\uparrow$ |
|---|---|---|---|---|---|
| Linear | 13.32 | 82.56 | 9.64 | 88.09 | 9.68 |
| DINOv2 (last layer) | 9.80 | 90.07 | 5.99 | 94.34 | 5.11 |
| DINOv3 (last layer) | 9.68 | 90.54 | 6.00 | 94.64 | 7.24 |
| DINOv3 (4 layers) | **9.29** | **91.18** | **5.54** | **95.08** | **13.47** |

from Figure 5, especially on object boundaries. This significantly limits the quality of diffusion output due to VAE bottleneck. Our method completely removes the VAE and

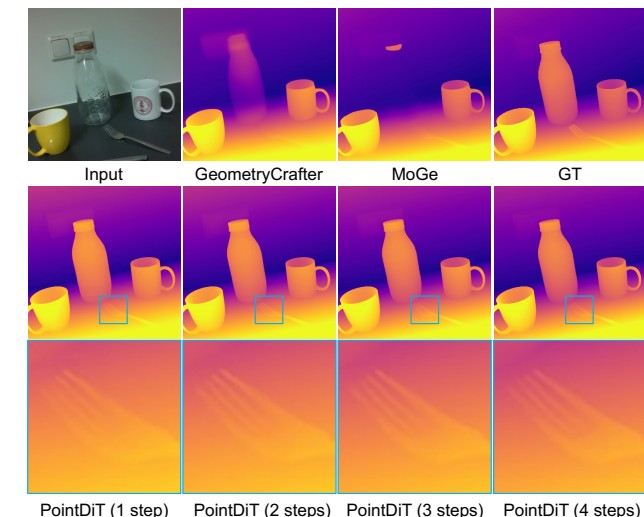

*Figure 4.* **Different diffusion sampling steps.** Our single-step diffusion already significantly outperforms prior works, and increasing the sampling steps further enhances reconstruction details (see the zoomed-in region).

*Table 6.* **Training loss**.

| Loss | $Rel^p \downarrow$ | $\delta_1^p \uparrow$ | $Rel^d \downarrow$ | $\delta_1^d \uparrow$ | BF1 $\uparrow$ |
|---|---|---|---|---|---|
| $v$-loss | 9.29 | 91.18 | 5.54 | 95.08 | 13.47 |
| $v$-loss & point loss | **9.10** | **91.48** | **5.53** | 94.88 | **13.92** |

*Table 7.* **Noise schedule**. Logit-normal and random zero $t$.

| $t$-shift | $Rel^p \downarrow$ | $\delta_1^p \uparrow$ | $Rel^d \downarrow$ | $\delta_1^d \uparrow$ | BF1 $\uparrow$ |
|---|---|---|---|---|---|
| $-0.8$ | 12.19 | 84.82 | 7.80 | 91.34 | **8.05** |
| $-1.2$ | 11.86 | 85.53 | 7.46 | 91.87 | 8.11 |
| $-1.6$ | 10.73 | 88.06 | 6.74 | 93.09 | 7.31 |
| $-0.8$ & rand zero | **9.68** | **90.54** | **6.00** | **94.64** | 7.24 |

*Table 8.* **Patch size**. Image resolution $512 \times 512$.

| Patch size | $Rel^p \downarrow$ | $\delta_1^p \uparrow$ | $Rel^d \downarrow$ | $\delta_1^d \uparrow$ | BF1 $\uparrow$ |
|---|---|---|---|---|---|
| 32 | 5.35 | 96.88 | 3.48 | 97.78 | 6.17 |
| 16 | **5.01** | **97.34** | **3.06** | **98.17** | **10.37** |

enables significantly more accurate local structures, as can be observed from Figure 3 and the BF1 metric in Table 1.

### 4.5. Ablation and Analysis

In this section, we evaluate the design choices of our model with controlled experiments by training on the $256 \times 256$ SceneNet-RGBD dataset and report the average metrics on the unseen test sets. For inference, we use all zeros for the noise sampling and thus the evaluation is done with deterministic single-step feed-forward inference.

**Prediction target: $x$-prediction vs. $v$-prediction.** Key to the success of our method is to predict the clean point

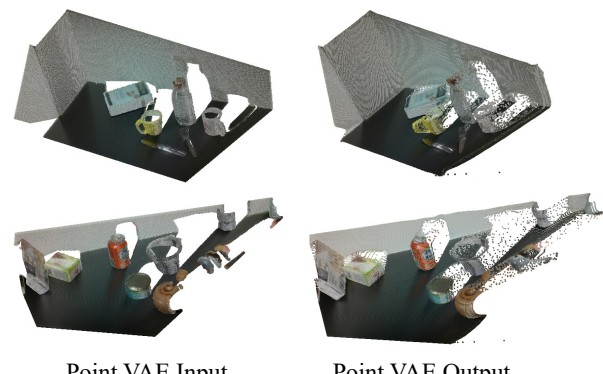

Point VAE Input      Point VAE Output

*Figure 5.* **Point cloud VAE produces significant artifacts.** Here we show the VAE reconstruction results from GeometryCrafter.

map ($x$-prediction) with the model instead of using the $v$-prediction. In Table 4, we can observe that the $v$-prediction fails catastrophically, which is consistent with the findings in JiT on image generation. In this paper, we demonstrate that the $x$-prediction is also crucial for point map diffusion.

**Noise schedule.** In the original JiT paper for image generation, a logit-normal noise scheduler is used. However, we observe that this alone leads to unsatisfactory results for point map generation where we measure per-point accuracy. In particular, since the timestep is obtained with the sigmoid function for the logit-normal sampler, it's nearly impossible to sample exact 0 during training. Thus, the model never sees pure noise during training, which potentially causes a train-test discrepancy and thus hurts the performance. Although this can be partially alleviated by shifting the noise schedule towards high noise regions with smaller mean values, the issue still exists. We propose to randomly set the sampled timestep to exact 0 with $10\%$ probability, which is a simple fix but significantly improves the quality.

**Diffusion *vs*. non-diffusion.** To further demonstrate the benefits of the diffusion formulation, we conduct another experiment by setting both the time step and the noise to always 0. This leads to a deterministic non-diffusion counterpart. As shown in Table 7, this leads to considerable performance drop, especially the BF1 boundary metric. This verifies the over-smoothness issue of the deterministic formulation in previous feed-forward models.

**Image patch embeddings.** In Table 5, we compare different patch embedding methods. Our model is able to achieve decent results without using any pre-trained image backbones. Our model can also benefit from pre-trained image patch embeddings. In particular, we evaluate using the last layer of DINOv2 and DINOv3 backbones and DINOv3 performs better. The results can be further improved by using 4 intermediate layers uniformly sampled from all layers, in particular the BF1 metric, indicating the effectiveness of

integrating different levels of abstractions from pre-trained image backbones.

**Training loss.** The ablation results discussed so far are obtained with the only denoising loss, which is already highly effective at capturing high-quality geometry. The results can be further boosted by using an additional relative loss (Equation (8)) that is specifically designed for the point data. As shown in Table 6, this loss brings additional gains.

**Patch size.** We further evaluate the impact of patch size for high-resolution images by fine-tuning the $256 \times 256$ pre-trained model to $512 \times 512$ resolution. We compare the patch size 16 and 32, and 16 leads to better overall results and sharper boundaries. This could be expected since the point map prediction tasks requires pixel-perfect accuracy, where a large patch size might lose the high-frequency details.

## 5. Conclusion

We presented a minimalist pixel-space diffusion model for monocular point map prediction that removes the architectural overhead of VAEs and hybrid networks. By validating that dense geometry can be modeled effectively in pixel space, we bridge the gap between standard image generation and 3D reconstruction. This paves the way for tokenizer-free 3D and 4D generation, relying solely on direct diffusion to model complex structural distributions.

**Limitation.** While our framework demonstrates robust geometric estimation, the current model is constrained by a fixed training resolution. Incorporating mixed-resolution training remains an important direction for future work to ensure seamless generalization across varying image resolutions. Furthermore, our empirical scaling analysis indicates that performance scales promisingly with dataset size. Thus, expanding the scale and diversity of the training data will likely be key to unlocking the model's full capabilities.

**Acknowledgements.** We thank Nando Metzger, Weirong Chen, Felix Wimbauer, Haiwen Huang, Gene Chou, Luca Zanella, Erik Sandström, Keisuke Tateno, Goutam Bhat, Mattia Segu, Vasile Lup, Tobias Fischer, Shaohui Liu and Bingxin Ke for the insightful discussions and support.

## Impact Statement

This paper presents work whose goal is to advance the field of machine learning and computer vision by simplifying the paradigm for monocular 3D reconstruction. There are some potential societal consequences of our work, ranging from advancements in robotics to spatial intelligence, none of which we feel must be specifically highlighted here from an ethical standpoint.

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
