# OpenReview forum: "PointDiT: Pixel-Space Diffusion for Monocular Geometry Estimation"
_ICML.cc/2026/Conference — ICML 2026 regular_

### Official Review · Reviewer_ANUc · 2026-02-26

**Soundness:** 4
**Presentation:** 4
**Significance:** 3
**Originality:** 3
**Overall Recommendation:** 4
**Confidence:** 4

**Summary:**

This paper tends to transfer the success of JiT in image generation to the monocular geometry estimation task. Technically, a pixel-space diffusion model is constructed based on the plain ViT architecture, with RGB image tokens extracted from frozen DINOv3 as conditions. The target output is the form of point map encoding spatial coordinates. Experiments demonstrate the effectiveness of the proposed framework.

**Compliance With Llm Reviewing Policy:**

Affirmed.

**Final Justification:**

Thanks for the authors' responses. My concerns have been basically addressed.

**Key Questions For Authors:**

NA

**Limitations:**

The proposed method itself shows relatively limited technical novelty. Basically, this paper only transfers the success of JiT to the different task of monocular geometry estimation. (Yet, it does not mean this work is not valuable.)

**Strengths And Weaknesses:**

### Strengths

This paper verifies the effective design choices proposed in JiT in the monocular geometry estimation task. It is valuable to demonstrate that pixel-space diffusion without complicated design also works well.

### Weaknesses

1. For image conditioning, the authors mention that while the input RGB image could be processed via standard learnable patch embeddings, they instead leverage powerful pre-trained representations from DINOv3. However, the corresponding quantitative verifications are missing.
2. It is claimed that using more inference steps could largely improve local structures. It is necessay to provide some visual comparisons.
3. Table 7 is unclear. Please add more necessary explanations to its settings.

---

> ### Author Rebuttal · Authors · 2026-03-30
>
> We thank the reviewer for the insightful comments.
>
> > The quantitative verifications of standard linear patch embeddings vs. DINOv3.
>
> We kindly refer the reviewer to Table 5 of our main paper for the quantitative verifications (also shown below).
>
> | Embedding | Rel$^p \\downarrow$ | $\\delta\_1^p \\uparrow$ | Rel$^d \\downarrow$ | $\\delta\_1^d \\uparrow$ | BF1 $\\uparrow$ |
> | :---- | :---: | :---: | :---: | :---: | :---: |
> | Linear | 13.32 | 82.56 | 9.64 | 88.09 | 9.68 |
> | DINOv3 | **9.29** | **91.18** | **5.54** | **95.08** | **13.47** |
>
> > Visual comparisons of more inference steps.
>
> We have provided visual comparisons across 1, 2, 3, and 4 sampling steps at [**anony1360.github.io**](http://anony1360.github.io) (Section C)**.** We can observe that PointDiT already achieves accurate results in a single step, and the sharpness gets further improved with more sampling steps. The model converges with 4 steps. We will include these visual comparisons in the revision.
>
> > More explanations to Table 7.
>
> We will expand the caption and main text for Table 7 to clarify the following experimental settings:
>
> - **Logit-Normal Shift:** This parameter controls the mean of the noise distribution $\\text{logit}(t) \\sim \\mathcal{N}(m, s^2)$. This is defined by transforming a Gaussian variable through the sigmoid function $t \= \\text{sigmoid}(x), \\quad x \\sim \\mathcal{N}(m, s^2)$, where $m$ is the Logit-Normal Shift and $s$ is the scale. A more negative shift (e.g., $-1.6$) biases training toward higher-noise regions (near $t=0$). Table 7 demonstrates that while increasing this shift generally improves performance, it reaches a plateau and requires hyperparameter tuning.
> - **Rand Zero ($p\_{\\text{zero}}$):** This denotes the probability of manually overriding the sampled timestep to $t=0$ during training. In standard logit-normal sampling, $t$ asymptotically approaches but never reaches the pure noise ($t=0$).
> - **Key Insight:** Table 7 shows that combining a moderate shift ($-0.8$) with Rand Zero ($p\_{\\text{zero}}=0.1$) yields the best performance across all metrics (e.g., $\\delta\_1^p \= 90.54$). This empirical result suggests that explicit boundary calibration at $t=0$ is effective for reducing train-test discrepancy.
>
> > The proposed method itself shows relatively limited technical novelty. Basically, this paper only transfers the success of JiT to the different task of monocular geometry estimation. (Yet, it does not mean this work is not valuable.)
>
> We respectfully clarify that our contribution extends beyond the straightforward "transfer" of JiT. While our PointDiT architecture deliberately adopts a minimalist Transformer philosophy, its success in monocular geometry estimation reveals a significant and non-obvious finding for the 3D community: hybrid architectures with complicated losses (e.g., MoGe) and VAE-based latent compression (e.g., GeometryCrafter) are fundamentally not required for high-fidelity 3D reconstruction.
>
> Furthermore, adapting a 2D image generation framework to raw 3D point maps is non-trivial.
>
> - **Task and Space Shift:** JiT is designed for class-conditioned generation in 2D image space. In contrast, our model performs image-conditioned diffusion directly in 3D point map space. Unlike bounded 2D RGB images, point maps are 3D coordinates which can have various ranges. We need proper normalization of the point maps to enable pixel-space diffusion to actually converge on raw geometry.
> - **Effective Conditioning via DINOv3:** Because our task is image-conditioned, we can uniquely leverage pre-trained DINOv3 features. This seamlessly transfers strong 2D priors to 3D structures.
> - **Noise Schedule:** JiT's noise sampling creates a train-test discrepancy since pure noise is never sampled during training. We fixed this issue by rectifying the noise schedule, ensuring the model is well-calibrated to the pure noise distribution encountered at the start of inference.
> - **Unprecedented Efficiency:** Standard diffusion models like JiT typically require 50+ sampling steps. As shown in our evaluations, our PointDiT achieves state-of-the-art results with single-step or few-step sampling, making our model significantly more efficient and practical for real-world applications.
>
> In summary, we view the streamlined nature of our architecture as a core strength rather than a limitation. We successfully provide a powerful, streamlined alternative to today's heavily engineered pipelines (e.g., GeometryCrafter). We will clarify this motivation in the revised manuscript.
>
> We hope these clarifications address the reviewer’s concerns and we remain open to further discussion.

---

### Official Review · Reviewer_fuH8 · 2026-03-10

**Soundness:** 3
**Presentation:** 3
**Significance:** 2
**Originality:** 2
**Overall Recommendation:** 4
**Confidence:** 4

**Summary:**

The authors presents PointDiT, a monocular image to depth map framework that utilizes DINO priors to do pixel-level depth estimation. The method is more streamlined compared to previous methods by removing the use of a VAE. The paper claims to achieve higher fidelity and accuracy in its empirical results, especially in traditionally more challenging regions.

**Compliance With Llm Reviewing Policy:**

Affirmed.

**Final Justification:**

The authors provide a good rebuttal and address most of my concerns, however I would still encourage the authors to include the talking points they mentioned in the rebuttal to the supplementary materials if appropriate.

**Key Questions For Authors:**

1. See Weakness #2
2. See Weakness #3
3. See Weakness #4

**Limitations:**

The authors mentioned briefly about fixing their generation dimension to 256x256, but without saying explicitly why this limitation exist. With the context of the paper, I assume this is a fundamental architectural limitation of replacing a latent VAE (which scales well) with a pixel-level generation paradigm (which is computationally heavy).

**Strengths And Weaknesses:**

### Strengths

- As the author stated, the method proposed is more streamlined compared to previous methods, which is a plus for reimplementation and reproduction of results.
- The preference on pretrained DINO priors over pretrained VAE appears to produce better results in reconstruction.
- Good results on inference even with one-step generation vs multi-step generation.

### Weaknesses

- The paper claims that their method is parameter-free that “immediately reconstructs 3D”, however, the model is trained on normalized data to satisfy the need of flow matching. As a result, inference will produce normalized data as well. In practice, this will need alignment from normalization parameters to end up with usable result.
- The proposed method is trained and fine-tuned on synthetic data, this might play a down factor in the presentation of this method as a well-generalized approach towards mono to 3D lifting. Many in the wild scenarios might see the results degrade under domain gaps for the current form of the paper’s models.
- The approach depends on a frozen DINOv3 backbone for conditioning. The paper does not clearly quantify the end-to-end compute/memory overhead versus latent-diffusion baselines, so the scalability/efficiency part of their narrative is incomplete. They mentioned in the experiment set up section that their “high-performance hardware configuration keeps fine-tuning times remarkably efficient”: this is not a positive talking point as they used 128 H100.
- The authors used a practical patch of randomly setting time step to 0 during training. This is somewhat of a volatile and heuristic approach and I’d like to see a proof of its effects.
- Fixed generation resolution might see this method less practical for real-world use cases, since the authors mentioned this in their discussion, this is less of a problem from my point of view. Overall, the method gives me a “replace previous pipeline with DINO” vibe, which is less impressive conceptually considering [1] already exists. If the authors can prove its practical values then this might enhance their narrative.

[1] Zheng, B., Ma, N., Tong, S., and Xie, S. Diffusion transformers with representation autoencoders. arXiv preprint arXiv:2510.11690, 2025.

---

> ### Author Rebuttal · Authors · 2026-03-30
>
> We thank the reviewer for the constructive feedback.
> > The reviewer’s summary: PointDiT, a monocular **image to depth map** framework for **depth estimation**.
>
> We first clarify that PointDiT is an **image-to-point-map** model, not merely an "image-to-depth-map" framework (ie, our model needs to learn camera intrinsics).
>
> > The normalized point map prediction will need alignment to end up with usable result.
>
> While recovering absolute metric space indeed requires solving for scale and shift alignment, our formulation follows the common practice of state-of-the-art baselines (e.g., MoGe), which also predict normalized point maps. Moreover, we respectfully clarify that unaligned, affine-invariant point maps are also usable in practice:
>
> - **3D Photography:** In computational photography (e.g., 3D Ken Burns \[Niklaus et al., 2019\]), affine-invariant geometry is sufficient for novel camera trajectories.
> - **Supervising 3D Optimization:** In sparse-view view synthesis and surface reconstruction, relative monocular geometry is the standard structural prior used to prevent artifacts (e.g., MonoSDF, Yu et al., 2022).
>
> > Generalization to real-world scenes.
>
> We kindly refer the reviewer to L260-L274 of our main paper due to the character limit. In short, PointDiT maps robust DINOv3 latent features to 3D point maps, thereby inheriting DINOv3's internet-scale pre-training. We also note that all the 7 evaluation datasets used in our paper are real-world datasets, and we provide **100+** visual results at [**anony1360.github.io**](http://anony1360.github.io) (Section A).
>
> > Efficiency vs. latent diffusion.
>
> We clarify that the use of 128 GPUs was simply a hardware availability choice to minimize wall-clock time (which only lasted 3 hours). The actual computational budget for this phase is roughly 384 GPU-hours. This is equivalent to training on a standard 8-GPU node for two days. We will revise the manuscript to avoid confusion.
>
> Compared to latent diffusion models, specifically Stable Diffusion 2.1, which requires an estimated 150,000 to 200,000 A100 GPU-hours, our pixel-space diffusion training is highly efficient. In terms of inference efficiency, our PointDiT-L already outperforms GeometryCrafter with **18x faster inference speed** and **2.5x fewer parameters** (Table 1).
>
> > The authors randomly set the time step to 0 during training.
>
> - **Technical Justification:** Standard timestep sampling ($\\sigma(t)$) asymptotically approaches but never reaches exactly 0\. This creates a train-test discrepancy, as inference begins at the pure noise ($t=0$). This issue has been discussed more thoroughly in \[Lin et al. Common Diffusion Noise Schedules and Sample Steps are Flawed. 2024\]. Explicitly setting $t=0$ for 10% of samples serves as a boundary calibration, ensuring the model learns the correct initial trajectory.
>
> - **Empirical Evidence:** In Table 7, we compared our "random $t=0$" approach against the standard method of shifting the logit-normal noise distribution. While increasing the shift (from $-0.8$ to $-1.6$) improves performance, our strategy ($p\_{\\text{zero}}=0.1$) consistently outperforms all shifted variants (e.g., $\\delta\_1^p$ improves from $88.06$ to $90.54$) without requiring sensitive hyperparameter tuning.
>
> > The authors fixed their generation dimension to 256x256.
>
> We first clarify that our PointDiT is not limited to a single 256x256 resolution; we evaluated on both 256x256 and 512x512, the standard resolutions for Diffusion Transformers. We have also trained a mixed-resolution variant and provided zero-shot results at [**anony1360.github.io**](http://anony1360.github.io) (Section B).
>
> Regarding the robustness to different resolutions, we attribute this to the inductive bias of ViT vs. CNN. CNN-based models use translation-invariant kernels, which adapt better to varying resolutions. In contrast, our PointDiT uses a plain ViT, which is more sensitive to the patch-grid shifts associated with varying resolutions. We are actively investigating techniques to further enhance robustness of PointDiT across arbitrary resolutions.
>
> > Practical values vs. RAE (Zheng et al.)
>
> - **End-to-end Training:** RAE needs to be trained in two stages (reconstruction decoder and diffusion), while PointDiT is end to end.
>
> - **Smaller Model Size:** RAE uses v-prediction, which requires scaling up the Transformer width (Section 4.1 of RAE paper), while PointDiT uses x-prediction and we successfully trained a smaller model variant PointDiT-B (width < latent dim).
>
> - **Practical Efficiency:** RAE requires 50 sampling steps for image synthesis, while PointDiT is able to perform one-step or few-step diffusion generation.
>
> Regarding the differences in task, coordinate space, noise schedule, and our minimalist paradigm shift, we kindly refer the reviewer to our detailed response to Reviewer 3ESy due to the character limit.
>
> We hope these clarifications address the reviewer’s concerns and we remain open to further discussion.

---

> > ### Author Rebuttal · Reviewer_fuH8 · 2026-04-03
> >
> > The authors provide a good rebuttal and address most of my concerns, however I would still encourage the authors to include the talking points they mentioned in the rebuttal to the supplementary materials if appropriate. I will rise my score to 4 (weak accept).

---

> > > ### Author Response · Authors · 2026-04-03
> > >
> > > Dear Reviewer,
> > >
> > > Thank you very much for your time, your constructive feedback throughout the review process, and for updating your score. We will absolutely ensure that the talking points and clarifications from our rebuttal are included in the supplementary materials of the revised manuscript.
> > >
> > > Furthermore, we sincerely appreciate the insightful comments from the entire review panel.  We are fully committed to incorporating the feedback and suggestions raised by all reviewers to further strengthen the final manuscript.
> > >
> > > Thank you again for helping us improve our paper.

---

### Official Review · Reviewer_3ESy · 2026-03-12

**Soundness:** 3
**Presentation:** 4
**Significance:** 4
**Originality:** 2
**Overall Recommendation:** 4
**Confidence:** 4

**Summary:**

This paper presents PointDiT, which is a pixel-space diffusion model that predicts dense point maps conditioned on image tokens from DINOv3. The authors showed that directly estimating the point map by x-prediction objective is effective for geometric data and eliminates the need for compressing point maps into a latent space. Experimental results show the robustness of the model on real-world datasets in various environments.

**Compliance With Llm Reviewing Policy:**

Affirmed.

**Final Justification:**

The authors have fully addressed my concerns and questions, and thus i would like to keep my score: weak accept

**Key Questions For Authors:**

- Since the model outputs affine-invariant point maps, how does the alignment procedure affect evaluation fairness?
- The denoising loss is computed in the velocity space while the relative point loss is applied directly to the predicted clean map. Is any training instability caused by these losses?

**Limitations:**

yes

**Strengths And Weaknesses:**

Strengths
1) The paper demonstrates an important shift towards 3D and 4D generation by proving the effectiveness of pixel-space diffusion for 3D tasks. The flexible inference on different steps also adds practical value.
2) The paper is structured and easy to follow, presenting the limitations of current works and motivations for the proposed work. The experimental results are clearly presented.

Weaknesses
1) Although the paper introduces task-specific techniques like point map normalization and relative point loss, the overall contribution shares significant overlap with JiT. The architectural design remains largely the same except for the addition of DINOv3 conditioning.
2) The ablation study for the training loss (Table 6) shows only marginal improvements across all metrics. It is unclear whether the additional relative loss provides meaningful impact on the model’s geometric reasoning.
3) The paper lacks sufficient explanation regarding the layer selection of the DINOv3 encoder.

---

> ### Author Rebuttal · Authors · 2026-03-30
>
> We thank the reviewer for the insightful comments.
>
> > Differences with JiT.
>
> We respectfully clarify that our contribution extends beyond a straightforward adaptation of JiT. While our PointDiT architecture deliberately adopts a minimalist Transformer philosophy, its success in monocular geometry estimation reveals a significant and non-obvious finding for the 3D community: hybrid architectures with complicated losses (e.g., MoGe) and VAE-based latent compression (e.g., GeometryCrafter) are fundamentally not required for high-fidelity 3D reconstruction.
>
> Furthermore, adapting a 2D image generation framework to raw 3D point maps is non-trivial.
>
> - **Task and Space Shift:** JiT is designed for class-conditioned generation in 2D image space. In contrast, our model performs image-conditioned diffusion directly in 3D point map space. Unlike bounded 2D RGB images, point maps are 3D coordinates which can have various ranges. We need proper normalization of the point maps to enable pixel-space diffusion to actually converge on raw geometry.
> - **Effective Conditioning via DINOv3:** Because our task is image-conditioned, we can uniquely leverage pre-trained DINOv3 features. This seamlessly transfers strong 2D priors to 3D structures.
> - **Noise Schedule:** JiT's noise sampling creates a train-test discrepancy since pure noise is never sampled during training. We fixed this issue by rectifying the noise schedule, ensuring the model is well-calibrated to the pure noise distribution encountered at the start of inference.
> - **Unprecedented Efficiency:** Standard diffusion models like JiT typically require 50+ sampling steps. As shown in our evaluations, PointDiT achieves state-of-the-art results with single-step or few-step sampling, making our model significantly more efficient and practical for real-world geometry estimation.
>
> In summary, we view the streamlined nature of our architecture as a core strength rather than a limitation. We successfully provide a powerful, streamlined alternative to today's heavily engineered pipelines (e.g., GeometryCrafter). We will clarify this motivation in the revised manuscript.
>
> > The impact of the additional relative loss.
>
> We appreciate the reviewer’s observation. Since our core pixel-space diffusion loss is already highly effective at capturing geometry, the room for auxiliary improvement appears to be relatively smaller. With the additional loss, we wanted to demonstrate PointDiT is a flexible framework that can easily incorporate additional geometric constraints when higher precision is required. This shows that practitioners can further refine the model's prediction using task-specific losses without compromising the core diffusion process.
>
> > The layer selection of the DINOv3 encoder.
>
> We thank the reviewer for the thoughtful comment. Our decision to extract DINOv3 features from four uniformly spaced layers directly follows the established standard for adapting Vision Transformers to dense prediction tasks, pioneered by DPT (Ranftl et al., 2021). By adhering to the common practice, we enable our model to extract a rich feature hierarchy. We will explicitly clarify this justification in the revised manuscript.
>
> > Since the model outputs affine-invariant point maps, how does the alignment procedure affect evaluation fairness?
>
> To ensure a strictly fair comparison, we use the exact same alignment procedure and evaluation code as MoGe, which is an established approach in this field. For aligning the affine-invariant point maps, the standard protocol involves solving for optimal scale and shift parameters to align predictions with the ground truth. By applying this identical mathematical alignment and using the same alignment code across different methods, we guarantee no model receives an unfair advantage from the fitting process. Thus, our reported gains strictly reflect superior geometric predictions.
>
> > The denoising loss is computed in the velocity space while the relative point loss is applied directly to the predicted clean map. Is any training instability caused by these losses?
>
> We did not observe any training instability. Because the estimated velocity $\\hat{\\mathbf{v}}\_t$ is a direct algebraic transformation of the predicted clean map $\\hat{\\mathbf{x}}$ (Eq. 6), gradients from both the denoising and relative point losses backpropagate seamlessly through the exact same $\\hat{\\mathbf{x}}$ output without conflict. Furthermore, we explicitly ensure numerical stability by clipping the $(1-t)$ denominator to prevent gradient explosion as $t \\to 1$.
>
> We hope these clarifications address the reviewer’s concerns and we remain open to further discussion.

---

> > ### Author Rebuttal · Reviewer_3ESy · 2026-04-04
> >
> > The authors have addressed my concerns and questions

---

### Official Review · Reviewer_9dSW · 2026-03-12

**Soundness:** 3
**Presentation:** 3
**Significance:** 3
**Originality:** 4
**Overall Recommendation:** 4
**Confidence:** 3

**Summary:**

The authors proposed a minimalist pixel-space Diffusion Transformer for monocular geometry estimation. By involving DINOv3 as a pre-trained tokenizer, the lightweight model can be trained from scratch in pixel space. The newly designed architecture achieves comparable results to those of complex latent-based diffusion models.

**Compliance With Llm Reviewing Policy:**

Affirmed.

**Final Justification:**

The additional experimental results and the explanation have addressed my concerns. I will keep my overall recommendation as "weak accept".

**Key Questions For Authors:**

1. In Table 1, how will it be when performing more steps for PointDiT-B/PointDiT-L? Will PointDiT-B or PointDiT-L still be comparable with MoGe with similar parameters?
2. How is the result of UniDepthV2 on the same evaluation datasets in Table 1?

**Limitations:**

yes.

**Strengths And Weaknesses:**

Strength:
1. The authors utilize the pre-trained DINOv3 as the tokenizer, eliminating the need for point map tokenizers.
2. The newly designed lightweight architecture still achieves comparable results with complex latent-based diffusion models, which is a simple yet effective design.
3. The experiments are conducted adequately to prove the effectiveness of the novel design, including prediction target, different image embeddings, noise schedule, and patch size.

Weakness:
1. From Table 1, although the PointDiT-H achieves great performance, the number of parameters is much more than models like Depth-Pro and MoGe-2.
2. The compared baselines include MoGe and DepthPro, but neglect other SOTA methods, like UnidepthV2. The visual comparisons are also limited to four images.

---

> ### Author Rebuttal · Authors · 2026-03-30
>
> We thank the reviewer for the constructive feedback.
>
> > Will PointDiT-B or PointDiT-L be comparable with MoGe with more diffusion steps?
>
> The results for PointDiT-B/L across multiple diffusion steps are provided in Table A.
>
> **Table A: Comparison with previous methods**
>
> | Method | $\\text{Rel}^\\text{p} \\downarrow$ | $\\delta\_1^\\text{p} \\uparrow$ | $\\text{Rel}^\\text{d} \\downarrow$ | $\\delta\_1^\\text{d} \\uparrow$ | $\\text{BF1} \\uparrow$ | Param (M) | Time (ms) |
> | :---- | :---: | :---: | :---: | :---: | :---: | :---: | :---: |
> | UniDepthV2 | 4.45 | 97.35 | 2.86 | 98.52 | 6.94 | 354 | 26 |
> | MoGe | **4.21** | 97.45 | 3.10 | 98.01 | 5.61 | 314 | 34 |
> | MoGe-2 | 4.53 | 97.46 | 2.90 | 98.45 | 7.40 | 326 | 24 |
> | PointDiT-B (1 step) | 5.84 | 96.71 | 3.70 | 97.84 | 8.18 | 223 | 31 |
> | PointDiT-B (2 steps) | 5.81 | 96.77 | 3.64 | 97.86 | 8.88 | 223 | 47 |
> | PointDiT-B (4 steps) | 5.85 | 96.80 | 3.64 | 97.86 | 9.16 | 223 | 79 |
> | PointDiT-L (1 step) | 4.90 | 97.42 | 3.15 | 98.22 | 9.56 | 771 | 65 |
> | PointDiT-L (2 steps) | 4.84 | 97.52 | 3.09 | 98.24 | 10.11 | 771 | 87 |
> | PointDiT-L (4 steps) | 4.85 | 97.55 | 3.09 | 98.25 | **10.49** | 771 | 131 |
> | PointDiT-H (4 steps) | 4.40 | **98.02** | **2.75** | **98.54** | **10.49** | 1807 | 204 |
>
> - **Parameter Efficiency:** PointDiT-B (223M) is smaller than MoGe-2 (326M), yet PointDiT-B (1-step) already achieves a superior BF1 score (8.18 vs. 7.40). This demonstrates that our pixel-space diffusion captures sharper geometric boundaries than hybrid architectures, even at a lower parameter budget.
> - **Scaling with More Steps:** With 4 steps, PointDiT-L achieves global metrics comparable to MoGe-2 while providing significantly better geometric sharpness (BF1 10.49 vs. 7.40).
> - **Controlled comparison:** Beyond model scaling, we want to emphasize that our performance gains are fundamentally driven by our diffusion-based formulation. As demonstrated in Table 3 of our main paper, we conducted a strictly controlled experiment to isolate this variable:
>
> | Method | $\\text{Rel}^\\text{p} \\downarrow$ | $\\delta\_1^\\text{p} \\uparrow$ | $\\text{Rel}^\\text{d} \\downarrow$ | $\\delta\_1^\\text{d} \\uparrow$ | $\\text{BF1} \\uparrow$ |
> | :---- | :---: | :---: | :---: | :---: | :---: |
> | Non-diffusion | 10.50 | 88.61 | 6.37 | 93.87 | 8.09 |
> | Diffusion | **9.10** | **91.48** | **5.53** | **94.88** | **13.92** |
>
>   In this experiment, both models share the exact same network architecture, parameter count, and training data. The "Non-diffusion" model is trained with a standard regression objective (conceptually similar to MoGe), while ours uses the diffusion objective. The results clearly show that the diffusion formulation significantly outperforms its regression counterpart across all metrics, with a particularly significant gain in structural sharpness (BF1). This confirms that our method's superiority is an inherent advantage of the generative paradigm, rather than simply a byproduct of larger model parameters.
>
> > How is the result of UniDepthV2 on the same evaluation datasets in Table 1?
>
> We thank the reviewer for highlighting UniDepthV2. As demonstrated in the new results in Table A, our method outperforms UniDepthV2 across all evaluated metrics. We will incorporate these UniDepthV2 results into the expanded evaluation tables in the revised manuscript.
>
> To clarify our original baseline selection strategy, our objective was to evaluate our approach against the strongest representative methods across different paradigms to ensure a rigorous comparison. We specifically selected our baselines as follows:
>
> - MoGe: The state-of-the-art representative for **regression**\-based models.
> - GeometryCrafter: The state-of-the-art representative for **diffusion**\-based models.
> - DepthPro: The leading method specifically focusing on **structural sharpness**.
>
> Because MoGe generally establishes a higher performance ceiling than UniDepthV2 for 3D point map prediction, we prioritized MoGe in the main text. However, we agree with the reviewer and will incorporate the UniDepthV2 results in Table 1.
>
> > The visual comparisons are limited.
>
> We agree that providing more visual examples is crucial for a comprehensive qualitative assessment. We initially limited the visual comparisons due to the page limits in the main paper.
>
> We have conducted an extended qualitative evaluation on over **100+** additional diverse scenes at [**anony1360.github.io**](http://anony1360.github.io) (Section A). Our method consistently yields better visual results than previous methods, demonstrating a particularly noticeable advantage in resolving thin structures and handling highly ambiguous regions like transparent and reflective surfaces. We will include these visual comparisons in our revised manuscript.
>
> We appreciate the time you took to help us improve our work, and we are more than happy to engage in further discussion should you have any remaining questions.

---

> > ### Author Rebuttal · Reviewer_9dSW · 2026-04-04
> >
> > The additional experimental results and the explanation have addressed my concerns.

---

### Decision · Program_Chairs · 2026-04-30

**Decision:**

Accept (regular)

**Comment:**

All reviewers recommend the weak accept in the first round review. All reviewers recognized the technical contributions of the paper. Reveiwers' concerns are well addressed in the rebuttal phase. Based on the comments of the paper, this paper is ready for ICML.